# Cardiac Resynchronization Therapy and Hypertrophic Cardiomyopathy: A Comprehensive Review

**DOI:** 10.3390/biomedicines11020350

**Published:** 2023-01-26

**Authors:** Andrei Dan Radu, Cosmin Cojocaru, Sebastian Onciul, Alina Scarlatescu, Alexandru Zlibut, Alexandrina Nastasa, Maria Dorobantu

**Affiliations:** 1Faculty of Medicine, “Carol Davila” University of Medicine and Pharmacy, 050474 Bucharest, Romania; radu_dan_andrei@yahoo.com (A.D.R.); cojocaru.r.b.cosmin@gmail.com (C.C.); sebastian.onciul@gmail.com (S.O.); mariadorobantu@yahoo.com (M.D.); 2Cardiology Department, Emergency Clinical Hospital of Bucharest, 014461 Bucharest, Romania; alina.scarlatescu@gmail.com; 3Department of Internal Medicine, Iuliu Hatieganu University of Medicine and Pharmacy, 400012 Cluj-Napoca, Romania; 4Cardiology Department, “Elias” University Emergency Hospital, 011461 Bucharest, Romania; alexandrina.nastasa@yahoo.com

**Keywords:** hypertrophic cardiomyopathy, end-stage/dilated hypertrophic cardiomyopathy, cardiac resynchronization therapy, left bundle branch block, apical rocking

## Abstract

Hypertrophic cardiomyopathy (HCM) is an inherited primary myocardial disease characterized by asymmetrical/symmetrical left ventricle (LV) hypertrophy, with or without LV outflow tract (LVOT) dynamic obstruction, and poor prognosis. Cardiac resynchronization therapy (CRT) has emerged as a minimally invasive tool for patients with heart failure (HF) with decreased LV ejection fraction (LVEF) and prolonged QRS duration of over 120 ms with or without left bundle branch block (LBBB). Several HCM patients are at risk of developing LBBB because of disease progression or secondary to septal myomectomy, while others might develop HF with decreased LVEF, alleged end-stage/dilated HCM, especially those with thin myofilament mutations. Several studies have shown that patients with myectomy-induced LBBB might benefit from left bundle branch pacing or CRT to relieve symptoms, improve exercise capacity, and increase LVEF. Otherwise, patients with end-stage/dilated HCM and prolonged QRS interval could gain from CRT in terms of NYHA class improvement, LV systolic performance increase and, to some degree, LV reverse remodeling. Moreover, several electrical and imaging parameters might aid proper selection and stratification of HCM patients to benefit from CRT. Nonetheless, current available data are scarce and further studies are still required to accurately clarify the view. This review reassesses the importance of CRT in patients with HCM based on current research by contrasting and contextualizing data from various published studies.

## 1. Background

Hypertrophic cardiomyopathy (HCM) is a primary myocardial disease characterized by asymmetrical left ventricle (LV) hypertrophy, with or without LV outflow tract (LVOT) dynamic obstruction and myocardial fibrosis, in the absence of ischemic heart disease and pressure overload conditions, such as valvular heart disease or arterial hypertension, which usually evolves with a poor prognosis [1]. Patients with HCM frequently develop heart failure (HF) phenomena mainly due to LVOT obstruction and LV diastolic dysfunction [2]; however, in up to 8% of cases, HCM can evolve with LV systolic dysfunction with decreased LV ejection fraction (LVEF)—alleged end-stage/dilated HCM [3]. 

Cardiac resynchronization therapy (CRT) has emerged as a useful minimally invasive tool in approaching patients with HF and reduced LVEF, and its effectiveness has been confirmed by various clinical trials [4]. CRT is usually recommended to patients with prolonged QRS duration of at least 120 ms and left bundle branch (LBB) block (LBBB); nevertheless, when QRS interval exceeds 150 ms, the latter becomes irrelevant [5]. 

What is more, patients with non-obstructive HCM might develop end-stage/dilated HCM, becoming possible candidates for LBB area pacing or CRT [6,7]. Besides, in those with obstructive HCM, septal myomectomy evolves with LBBB and enlarged QRS interval in around 40% of cases [8]. Electrical parameters, namely the QRS morphology, such as LBBB or nonspecific intraventricular conduction delay (NICD), and duration have proved to highly predict positive response of patients to CRT [9]. Also, there are various imaging parameters, such as LV “apical rocking”, “septal flash” and many more, which might aid the correct stratification of patients as candidates for CRT [10]. 

Hitherto, the role of implantable cardiac devices has been confirmed by several studies. Patients with obstructive HCM benefit from dual-chamber pacemakers [11], due to relief of symptoms and increase in their life-quality [12]. Furthermore, the same category of patients who underwent septal myomectomy has been shown to positively benefit from left bundle branch area pacing and CRT [7,13]. Moreover, in patients with end-stage/dilated HCM, CRT proved to significantly improve NYHA class, LVEF, LV volumes and cardiac output, also promoting LV reverse remodeling [14]. Nevertheless, future studies are required to enlighten our understanding.

The aim of this review is to provide a comprehensive overview of current available data about the role of CRT in patients with HCM.

## 2. HCM in a Nutshell

Hypertrophic cardiomyopathy (HCM) is the second most frequent inherited myocardial disease which requires comprehensive evaluation and risk stratification. Current therapeutic options have significantly lowered the morbidity and mortality of these patients; thus, active screening should become compulsory. HCM is an autosomal dominant inherited illness which has a higher prevalence in males, while in female patients, the occurrence of secondary heart failure (HF) is more common [1]. In over two thirds of cases the mutations in beta-myosin heavy chain-7 (MYH7) and myosin-binding protein C 3 (MYBPC3) represent the culprit genetic alterations which lead to the development of this disease, determining myocardial cell hypertrophy, disarray distribution and fibrosis [15]. Nevertheless, many other mutations have been shown to be responsible for up to one third of HCM cases [16].

These various genetic mutations and polymorphisms are accountable for HCM phenotypes through various molecular mechanisms, comprising (1) alterations of transcription and translation processes which lead to afflicted protein synthesis; (2) missense mutations that determine the synthesis of mutant proteins which are, in turn, incorporated into myocardial sarcomeres; (3) thick myofilament mutations which give rise to altered actomyosin complex, calcium ions sensitivity, and Adenosine 5’-Tri-Phosphatase (ATPase) activity; (4) thin myofilament mutations which increase calcium ions sensitivity and ATPase activity [15]. Therefore, various pathogenetic pathways are altered, leading to flawed functioning of calcineurin, mitogen-activated protein kinases, transforming growth factor β, noncoding ribonucleic acids (RNAs) and epigenetic factors [17,18,19,20,21,22].

It is well known that molecular myocardial derangements are responsible for the pathological manifestation of the HCM via myocardial hypertrophy, disarray and fibrosis [15]. Nevertheless, the clinical phenotypic variability has been proved to be directly linked to the causal gene mutation and, thus, the paramount role of genetic testing in patients with HCM. As part of personalized medicine, the accurate characterization of genetic mutations might aid therapy guidance and prognosis prediction [2]. If one refers to the most common gene mutations in HCM, which involves MYH7 and MYBPC3, clinical manifestations are mostly similar, resulting in myocardial hypertrophy, which can also be enhanced by increased LV loading pressures, such as high blood pressure or isometric exertion [23]. Moreover, thin myofilament mutations, such as cardiac troponin T, give rise to a moderate myocardial hypertrophy and a higher chance of LV global dysfunction [15,23]. Nonetheless, the risk of sudden cardiac death by malignant tachyarrhythmias does not differ significantly between causative genes [15]. In HCM, myocardial hypertrophy occurs mostly in the inferior and basal segments of the interventricular septum, involving LVOT, while other LV locations comprise the apex and, rarely, the lateral and posterior walls [1,2]. In many cases, myocardial hypertrophy is accompanied by mitral valve phenotypic alterations, mainly leaflet elongation and anomalous insertion of the correspondent papillary muscle [24]. 

Concerning LV malfunction, several clinical hallmarks are often present in patients with HCM: (1) increased LV systolic function due to abnormally reduced LV end-systolic volume; (2) LVOT dynamic obstruction at rest or exertion due to septal asymmetrical hypertrophy; (3) systolic anterior movement of the mitral valve due to phenotypic alterations of the mitral valve or its apparatus; (4) LV diastolic dysfunction with increased myocardial stiffness due to cardiac diffuse and interstitial fibrosis [1,25]. 

## 3. Clinical and Imaging Approach in HCM

Usually, HCM becomes clinically overt when the pathogenetic alterations are sufficient to destabilize the heart, leading to angina, shortness of breath, palpitations, or syncope, while resting electrocardiographic alterations are closely related to the degree of myocardial hypertrophy; for instance, isolated interventricular septal hypertrophy determines long Q waves in the inferior-lateral leads, while in apical HCM, massive T waves in the precordial leads have been found [2]. As it is well known, HCM is strongly associated with both atrial and ventricular tachyarrhythmias. Atrial fibrillation often occurs in up to one quarter of patients with HCM and has a significant hemodynamical impact on these patients. Ventricular tachyarrhythmias comprise premature ventricular beats, non-sustained and sustained ventricular tachycardia and are responsible for a mortality rate of 0.5–1% per year [26]. 

On the other hand, advanced multimodal imaging plays a paramount role in diagnosing and risk stratifying of these patients, especially by the joint use of echocardiography and cardiac magnetic resonance imaging (CMR) [27]. Both European Society of Cardiology (ESC) and American Heart Association/American College of Cardiology (AHA/ACC) guidelines recommend standard transthoracic echocardiography (Figure 1) as the first-encounter imaging tool in patients suspected of HCM because it can provide a significant overview about LV hypertrophy, LV systolic and diastolic function, LVOT or intraventricular dynamic obstruction and mitral valve phenotypic particularities [28,29,30]. Echocardiographic findings in patients with HCM are described in Table 1 [27,28,31]. 

Nonetheless, due to increased temporal resolution, ability to comprehensively evaluate all heart structures and capacity to characterize myocardial tissue and unravel fibrosis, CMR has currently became the gold-standard imaging tool for diagnosing and assessing patients with HCM. CMR is able to accurately quantify heart chambers’ volumes and dimensions and also both LV and right ventricle (RV) systolic functions. Moreover, CMR is the only large-scale available noninvasive imaging tool capable of tissue characterization. This aspect has a tremendous role in patients with HCM since it can identify and accurately quantify both myocardial replacement and diffuse fibrosis and it can also exclude other infiltrative cardiomyopathies such as amyloidosis [27]. CMR summarized in Table 1. Specific cases of HCM assessed by CMR are presented in Figure 2, Figure 3 and Figure 4.

What is more, due to its increased risk of sudden cardiac death, over time, various HCM risk factors and scores which aimed to ease the identification of patients at higher risk of death have emerged. Amongst all, these comprise previous ventricular tachyarrhythmias, SCD in 1^st^-degree relatives, recurrent syncope due to cardiac arrhythmias, colossal myocardial hypertrophy of over 30 mm, LV aneurysm and “burn-out” HCM with decreased LV ejection fraction [26]. The AHA/ACC guidelines strongly recommend SCD risk assessment at initial evaluation and every 1-2 years at follow-up using the following parameters: family history of sudden cardiac death attributable to HCM, massive LV hypertrophy, unexplained syncope, end-stage/dilated HCM, LV apical aneurysm, extensive fibrosis quantified as LGE at CMR and nonsustained ventricular tachycardia [29].

## 4. Current Advances in CRT

CRT has emerged as a useful minimal invasive tool in approaching patients with HF and reduced LVEF, and its effectiveness has been confirmed by numerous clinical trials [4]. These clinical trials deployed significant results which strengthen the role of CRT in cardiac patients, by lowering mortality and morbidity, increasing LV systolic performance, enhancing LV reverse remodeling and decreasing the cardiovascular outcome in cardiac patients with enlarged QRS intervals. These trials are presented in Table 2 [32,33,34,35,36,37,38,39,40,41,42,43,44]. 

Nonetheless, in patients with narrow QRS complex, existent clinical trials did not provide any beneficial impact [45,46]. The main category of patients who benefit from biventricular pacing are those with moderate-to-severe HF and QRS complex duration of at least 120 ms, mostly serving patients with ischemic heart disease or non-ischemic dilated cardiomyopathy [4]. Hence, the latest ESC and ACC/AHA guidelines for cardiac pacing and CRT recommends heart resynchronization therapy to increase cardiac function, ameliorate symptoms and reduce morbi-mortality. The highest class and level of evidence (IA) stands for symptomatic HF patients in sinus rhythm with a LVEF of under 35% and QRS morphology of LBBB with duration of over 150 ms, while for those with a duration in between 130–149 msec the class (IIa) and level of evidence (B) are significantly lower and comparable to those with non-LBBB morphology [5] (Figure 3). 

Future research should be keen to improve clinical and prognostic response to CRT and to attempt to extend its current indications. Nearly half of patients who receive CRT later turn out to be non-responders to it. Standard echocardiography, the gold-standard imaging tool in approaching patients with HF, can be used to improve the selection of these patients [9]. The most relevant echocardiographic parameters for intraventricular dyssynchrony are LV “apical rocking” and “septal flash”, being also recommended by both ESC and ACC/AHA guidelines [5,47]. Moreover, using advanced echocardiography techniques, several parameters of intraventricular, interventricular and atrioventricular dyssynchrony have been reported; however, their utility is still questionable due to lack of sufficient data [48].

## 5. Distinguishing Patients with HCM Who Might Benefit from CRT

### 5.1. Electrical Parameters in Stratifying Patients with HCM for CRT

As HCM progresses, the myocardium becomes more fibrotic and electrically heterogenous, which, in some patients, leads to electrical dyssynchrony. This affliction severely impacts heart hemodynamics and concurs to HF phenomena and increases mortality [49]. The main target of CRT is represented by electrical dyssynchrony, which is considered to be the first event in what is called “dyssynchronopathies”. It endorses myocardial dysfunction and remodeling, substantially impacting LV systolic and diastolic functions [9]. Moreover, proper evaluation of patients with HCM to attempt to quantify the electro-mechanical dyssynchrony in order to increase their probability to become CRT responders has become imperative, for the reason that CRT serves as a biventricular pacing which, in the absence of dyssynchrony, in fact determines asynchronicity and worsens their prognosis [50]. Yet, the presence of HCM with ventricular dyssynchrony is not enough to stand as a recommendation for biventricular pacing, but they should always be accompanied by prolonged QRS interval of at least 120 ms. What is more, two clinical trials, namely LESSER-EARTH and ECHO-CRT, that followed the impact of CRT in patients with narrow QRS interval have been prematurely ended because they increased morbi-mortality [45,51].

Current international guidelines recommend standard surface ECG with QRS duration measurement for establishing the indication of CRT, regardless of patients’ disease. Nevertheless, the major shortcoming in surface ECG is that it cannot accurately characterize various myocardial activation irregularities and it presents with a wide variety in defining LBBB, both leading to a significant number of patients who are CRT non-responders [9]. Additionally, the inhomogeneity in identifying and categorizing LBBB is caused by gaps in its proper definition, guidelines’ unevenness and intra- and inter-observer variability [52,53]. Hence, in a meta-analysis conducted by Cleland et al., which evaluated 3782 patients with proper recommendation for CRT, amongst QRS duration, LVEF and LBBB morphology, solely QRS interval was independently associated with all-cause mortality and hospitalization for HF [53]. Moreover, patients who exhibit NICD are also candidates for CRT, mainly based on their QRS duration; however, there is a considerable inconsistency in their response rate, especially due to the underlying myocardial illnesses [54]. More than that, in a study conducted by Kawata et al., which evaluated the impact of CRT in patients with NICD and right bundle branch block, only QRS duration of over 150 ms was associated with a positive response rate [55]. Thus, in patients with HCM, not solely the LBBB morphology should determine CRT, but rather the QRS duration should also be always taken into account. Various ECG morphologies can be found in Figure 5.

Furthermore, another relevant matter is represented by patients with HCM who undergo septal reduction procedures, such as percutaneous transluminal septal myocardial ablation or surgical myomectomy. In the study of Qin et al., which evaluated 204 patients who underwent such procedures, the majority of those with surgical myomectomy developed LBBB with an average QRS duration of 150 ms ± 20 ms [56]. Likewise, in another recently published study on 2482 patients with obstructive HCM who underwent myomectomy, around 38.8% of patients developed LBBB, but did not incrementally influence their outcome [8]. In addition, Gionfriddo et al. have shown that after extensive septal myomectomy, the QRS duration significantly increased from 94 ms ± 10 ms to 152 ms ± 15 ms, along with LV end-diastolic diameter (from 40 ms ± 5.6 ms to 46.2 ms ± 6.5 ms) at two months after the procedures [57]. Although things are just at the beginning, CRT and other biventricular pacing methods might become useful in these patients.

### 5.2. Imaging Parameters in Stratifying Patients with HCM for CRT

Seldom, due to various reasons, the decision to perform CRT becomes inaccurate and incremental parameters which might predict a favorable clinical response are often required. Therefore, current technological advancements have deployed various imaging modalities that might provide useful measurements which might aid the judgement of implanting CRT devices. 

Echocardiography is the main imaging tool for evaluating cardiac patients, being highly available, and, thus, parameters for evaluating LV dyssynchrony based on this technique have continuously emerged. According to current available guidelines, LVEF is one of the main parameters used for recommending biventricular pacing [5]. Moreover, Marstrand et al. collected data from the SHaRe Registry and showed that around 8% of HCM patients develop end-stage/dilated HCM, having significantly increased risk for cardiovascular adverse events [3]. LVEF remains the key element in identifying patients who could benefit from biventricular pacing; however, other parameters offer promising results, although further studies need to be conducted [58]. 

In terms of CRT guidance, echocardiography can also be used to determine dyssynchrony parameters. LV “apical rocking” and “septal flash” represent the main predictive elements that should be assessed in candidates for biventricular pacing. “Apical rocking” has been shown to be independently associated with positive clinical and echocardiographic response to CRT [59]. The PREDICT-CRT trial, which sought to evaluate the relationship between “apical rocking” and “septal flash”, LV reverse remodeling and long-term survival in patients who underwent CRT, included 1060 patients. The trial proved that both parameters were independently associated with increased survival and added incremental predictive value for CRT response rate beyond QRS duration, while their absence was correlated with increased mortality and lower response [10]. 

Furthermore, various echocardiographic-based dyssynchrony parameters have emerged; however, current guidelines do not recommend them for standard clinical use due to their debatable efficacy [5]. As recently summarized by Mele et al., special echocardiography techniques can be used to assess multiple asynchrony parameters. Color tissue Doppler imaging echocardiography can be used to determine septal to lateral and to posterior wall delays, standard deviation of time-to-peak velocities and strain, overall exceeding time, while two-dimensional speckle-tracking echocardiography can determine strain delay index, longitudinal and transverse dyssynchrony indexes, LV torsion and twist, longitudinal and radial global dyssynchrony index, or radial discoordination index [48].

## 6. Current Prospects of CRT in Patients with HCM

Patients with HCM encounter progressive heart deterioration, especially due to disease progression, which genuinely leads to HF, impaired LV systolic function, enlarged heart chambers and electrical abnormalities. Conduction disorders are oftentimes approached using cardiac implantable electrical devices. A recently published study that sought to characterize patients with HCM who initially received a device concluded that implantable cardioverter defibrillators were the most common of them all, but neither dual-chamber pacemakers nor CRT were to be overlooked [11].

For the last two decades, growing evidence has shown the advantages of cardiac pacing in patients with HCM. Kappenberger et al. have initially proved the positive impact of standard pacing in patients with obstructive HCM, relieving symptoms, improving exertion and life-quality [12]. The same research team has subsequently shown that dual-chamber pacing was able to significantly reduce LVOT gradients from 72 ± 35 mmHg to 29 ± 24 mmHg [60]. Furthermore, it has been shown that dual-chamber pacing exceedingly maintained normal LVOT gradients, symptoms and exertion both at long-term and very-long-term follow-ups [61,62]. On the contrary, Nishimura et al. have found conflicting results: that in some HCM patients, dual-chamber pacing might relieve symptoms, while in others it worsened them [63]. Similarly, another study conducted on 48 obstructive HCM patients concluded that symptom relief was rather a placebo effect, reduction in LVOT gradient was trivial and cardiac pacing was not associated with meaningful improvement in cardiac function [64]. These contrasting results remain a matter of debate and further studies on larger cohorts of obstructive HCM patients are required to enlighten the view.

Some patients with HCM who survive in the long term develop HF with decreased LVEF and dilated LV, developing overlapping phenotypes of dilated cardiomyopathy—the soi-disant “burn-out stage” or end-stage/dilated HCM [65]. Hence, CRT might become a useful tool for this category of patients who also present with prolonged QRS. Furthermore, there are some HCM patients who might also develop sinus node dysfunction and require artificial permanent pacing; however, standard RV apical pacing increases the risk of interventricular dyssynchrony, HF, mitral regurgitation and atrial fibrillation [47]. Recently, Patra et al. presented a case report regarding a patient with non-obstructive HCM who developed sick sinus syndrome and received LBB area pacing and obtained excellent response to it at follow-up, maintaining intraventricular synchronicity, relieving symptoms and improving echocardiographic measurements [47]. Similarly, Zhang et al. performed a similar LBB area pacing in a pediatric patient with non-obstructive HCM and restrictive phenotype who presented with complete atrio-ventricular block and also reported favorable results [6]. In a recently published small, yet essential study, Zheng et al. aimed to evaluate the utility of LBB area pacing and His bundle pacing in 10 patients with HCM and post-myomectomy LBBB with standard recommendation for CRT. They found that the block-site was infranodal so that His bundle pacing was inefficient, while LBB area pacing was able to significantly narrow the QRS duration from 163.3 ± 16.6 ms to 123.6 ± 15.8 ms [7]. Additionally, in terms of CRT after septal myomectomy-induced LBBB in patients with obstructive HCM, a case report regarding a female patient with LBBB and refractory HF symptoms after myomectomy for obstructive HCM stated improved symptoms and life quality at 3 months after CRT implantation [13].

Several lately reported data endorse the use of CRT in patients with HCM, proving beneficial short- and long-term effects. In a case-control study, Arregle et al. sought to assess the impact of CRT on patients with HCM, decreased LVEF and HF in comparison with patients with DCM. At a median follow-up of 41 months, there were statistically significant improvements in terms of NYHA class, LVEF, LV volumes and cardiac output, thus presenting beneficial effects on symptoms and LV reverse remodeling [14]. In a recently issued meta-analysis which comprised 65 patients with HCM who received CRT, it has been reported significantly improved NYHA classes in these patients, even if LVEF and LV end-diastolic diameter remained unchanged [66]. 

Withal, a recently published study which aimed to compare the effects of CRT in patients with end-stage/dilated HCM by comparing them to patients with dilated cardiomyopathy and ischemic heart disease, pursued the occurrence of LV end-systolic volume decrease at 6-months follow-up. The study concluded that over half of patients with end-stage/dilated HCM were responders to CRT, with improved NYHA class, LVEF and LV end-systolic volume, while QRS duration narrowed from 158.7 ± 32.2 ms to 141.7 ± 22.0 ms, although they had a lower positive response rate to CRT than others (56.3% vs. 69.2% vs. 73.6%) [67]. Likewise, Nakajima et al. compared the impact of CRT on 22 patients with end-stage/dilated HCM and 71 with dilated cardiomyopathy. They have shown that in both groups, CRT significantly relieved symptoms; however, the effects on increasing LVEF and reducing LV end-systolic volume were lower than in those with dilated cardiomyopathy. Moreover, CRT was associated with similar survival and HF hospitalization at 2-year follow-up in both groups, but afterwards, the end-stage/dilated HCM group had significant disease progression and increased mortality [68]. Conversely, in a small study conducted on patients with end-stage/dilated HCM, Killu et al. have shown that in this category of subjects, CRT has rather debatable consequences. Initially, although CRT increased their LV systolic function, in the long term it did not persist, being similar to those without biventricular pacing. Also, they reported no beneficial effect on disease progression [69]. These contradictory results might be argued by variable “molecular responses” and further advanced tissue characterization before CRT might be useful in patients with HCM. Nonetheless, an algorithm regarding cardiac resynchronization in primary prevention of SCD in patients with HCM is presented in Figure 6.

## 7. Future Perspectives

Even though HCMs have similar pathogenetic background, due to various genetic mutations the phenotypic expression along with the direction of disease progression is quite variable. There are some mutations, such as of thin myofilaments, which are more likely to evolve with end-stage/dilated HCM [15], and further studies should enlighten if this category of patients might benefit more from CRT and would require closer monitoring in this direction. Furthermore, as has been shown by several studies, patients with end-stage/dilated HCM are identified late in their disease course when their LVEF is severely decreased and, thus, have variable responses to CRT. Some patients do not respond at all, while others present with symptom relief, increased exertion capacity and, to some extent, a more stable outcome ([67,69]). Nevertheless, all published studies have shown that disease progression was not very influenced by CRT. New inquiries should try to identify plausible explanation for these.

Another challenging facet regarding CRT in patients with HCM would be the correct selection of the device, either a simple pacing or with defibrillator (CRT-D). Certainly, for patients with sudden cardiac death, ventricular fibrillation or ventricular tachycardia, the current AHA/ACC guideline undoubtedly recommends implantable cardioverter defibrillator as secondary prevention [29]. However, the general risk of sudden cardiac death in patients with HCM is around 0.5%/year and does not clearly argue for implantable cardioverter defibrillator [70]. As for primary prevention, current AHA/ACC guidelines recommend establishing clinical risk factors for HCM sudden death risk stratification [29]. Further research should be conducted in this direction. More than that, supplementary attention should be given to patients with obstructive HCM who underwent septal myomectomy. In the prospective study of Woo et al. [71], which evaluated over 330 patients with HCM and myectomy, at follow-up the incidence of sudden cardiac death was under 4%, while in another similar study conducted by Ommen et al. [72], the 10-year survival was around 99% in those who underwent myectomy. Thus, HCM patients who undergo myectomy have a significantly decreased risk of sudden cardiac death, but it is not null. Those who require CRT should also be stratified according to current international guidelines in order to decide if a CRT-D would be a better option for them.

What is more, it seems that CRT-D with His bundle pacing might incrementally aid patients with HCM. Besides protection against ventricular arrhythmias, right ventricle captures might be avoided and, thus, cardiac dyssynchrony prevented. Moreover, some devices allow merely left ventricular pacing and selective His bundle pacing with physiological electrical activation of the heart [73]. Nonetheless, further studies need to be conducted to objectify the benefit.

## 8. Conclusion

In patients with HCM, disease progression might lead to HF with decreased LV systolic function and LBBB. Also, LBBB might be secondary to septal myomectomy. Thus, for these patients, CRT might become a useful therapeutic option and electrical and imaging parameters might aid their stratification and selection. Larger cohort studies are required to confirm these findings.

## Figures and Tables

**Figure 1 biomedicines-11-00350-f001:**
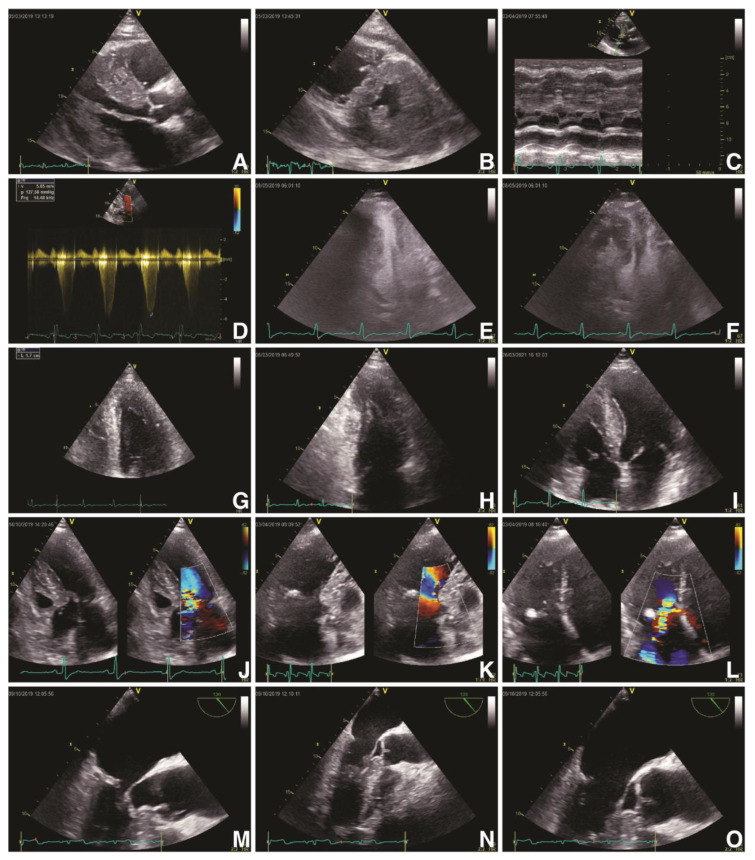
Echocardiographic findings in hypertrophic cardiomyopathy. TTE: obstructive HCM (Maron type III) in PLAX (**A**) and PSAX (**B**) views with an IVS of 23 mm and a ILW of 19 mm, with SAM of MV’s anterior leaflet in PLAX M-Mode (**C**) and a LVOT gradient of 127 mmHg and velocity of 5.65 m/s (**D**); contrast transthoracic echocardiography of a HCM with mid-cavity obstruction and apical LV aneurysm in A4C in systole (**E**) and diastole (**F**); HCM (Maron type IV) with isolated hypertrophy (18 mm) of the apical segment of LV’s lateral wall (**G**, **H**); non-obstructive HCM (Maron type II) with isolated hypertrophy of the IVS (20 mm) (**I**); obstructive HCM with SAM of the MV’s anterior leaflet and of its chordae tendineae with leads to severe MR (**J**–**L**). TEE: obstructive HCM with SAM of MV’s anterior leaflet—A2 scallop (**M**–**O**). Abbreviations: A4C, apical four-chamber view; HCM, hypertrophic cardiomyopathy; IVS, interventricular septum; LV, left ventricle; LVOT, left ventricle outflow tract; MV, mitral valve; PLAX, parasternal long-axis view; PSAX, parasternal short-axis view; SAM, systolic anterior movement; TEE, transoesophageal echocardiography; TTE, transthoracic echocardiography.

**Figure 2 biomedicines-11-00350-f002:**
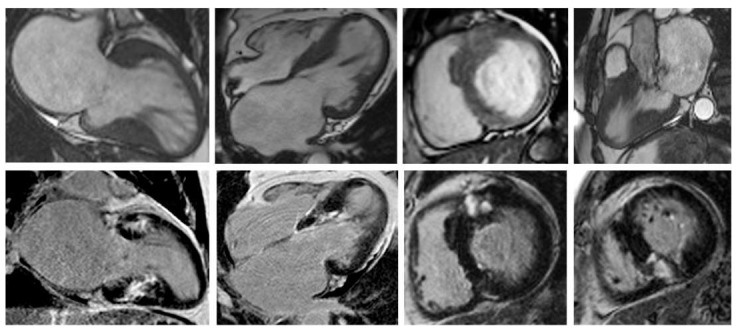
Contrast-enhanced CMR in a 50-year-old male patient with hypertrophic cardiomyopathy (from the own collection of the authors). (**Upper row**): b-SSFP cine images in two chambers, four chambers, short axis and three chambers, respectively. There is asymmetrical LV hypertrophy, with a maximum thickness of 20 mm at the basal anterior wall. The LV is not dilated and the LVEF is within normal limits (59%). There is systolic anterior motion of the mitral valve, resulting in acceleration of the systolic flow in the LV outflow tract. (**Lower row**): LGE imaging shows patchy myocardial fibrosis of the hypertrophied segments. Abbreviations: b-SSFP, balanced steady-state free precession; CMR, cardiac magnetic resonance imaging; LVEF, left ventricle ejection fraction; LGE, late gadolinium enhancement; LV, left ventricle.

**Figure 3 biomedicines-11-00350-f003:**
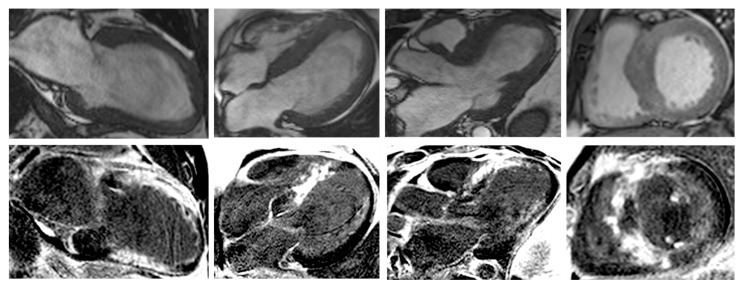
Contrast-enhanced cardiovascular magnetic resonance imaging in a 32-year-old male patient with a history of hypertrophic cardiomyopathy due to a pathogenic mutation in MYBPC3 gene (c.772G>A) (from the own collection of the authors). The current findings are compatible with a phenotype of burn-out HCM. (**Upper row**): b-SSFP cine images in two, four and three chambers, and short axis, respectively. The LV is severely dilated (166 mL/m2) with severe systolic dysfunction (LVEF 22%). Currently, there is no LV outflow tract obstruction, although an obstructive phenotype was diagnosed 10 years before. (**Lower row**): LGE imaging shows extensive heterogeneous replacement myocardial fibrosis at the level of the interventricular septum, anterior and inferior LV walls, respectively. Also note the fibrosis of the papillary mitral muscles. Abbreviations: b-SSFP, balanced steady-state free precession; CMR, cardiac magnetic resonance imaging; LVEF, left ventricle ejection fraction; LGE, late gadolinium enhancement; LV, left ventricle.

**Figure 4 biomedicines-11-00350-f004:**
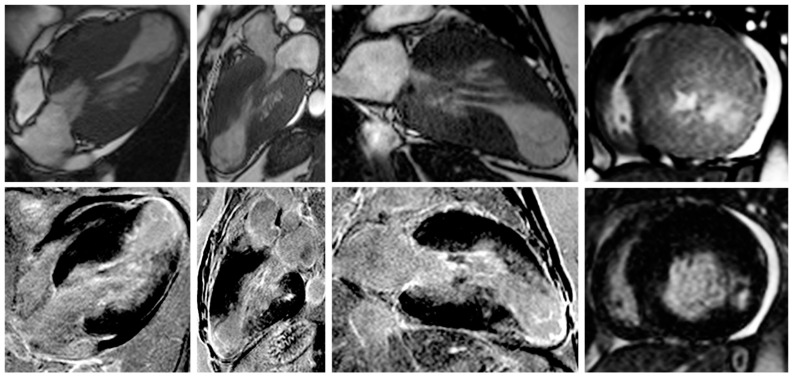
Contrast-enhanced CMR in a 50-year-old female patient with a history of hypertrophic cardiomyopathy with LV apical aneurysm (from the own collection of the authors). (**Upper row**): b-SSFP cine images in four, three and two chambers, and short axis, respectively. The LV is dilated (110 mL/m^2^) with severely impaired systolic function (LVEF 25%). There is asymmetrical LV hypertrophy with a maximum wall thickness of 28 mm at the level of the interventricular septum. Note the large apical aneurysm with a maximum diameter of 40 mm and thin walls. (**Lower row**): LGE imaging shows transmural fibrosis of the LV apical aneurysm and a small focal intramyocardial scar at the level of the basal infero-lateral wall. The total percent of fibrosis is 16% of the LV myocardium. No thrombus is seen inside the apical aneurysm. Abbreviations: b-SSFP, balanced steady-state free precession; CMR, cardiac magnetic resonance imaging; LVEF, left ventricle ejection fraction; LGE, late gadolinium enhancement; LV, left ventricle.

**Figure 5 biomedicines-11-00350-f005:**
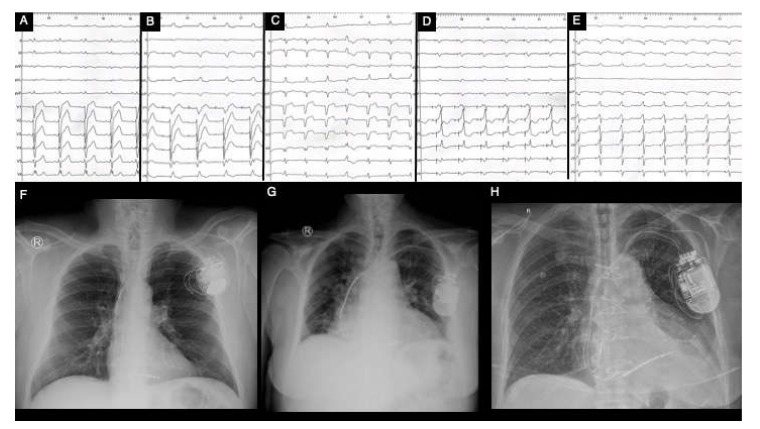
Clinical vignette of LBBB and post-CRT electrical patterns and post-procedural radiological anatomy in patients with hypertrophic cardiomyopathy treated with CRT (own collection of authors). (**A**)—typical LBBB activation; (**B**)—prior AAIR pacing for sinus node disease with atypical LBBB activation; (**C**)—atypical LBBB with apical aneurysm in midventricular hypertrophy pattern HCM; (**D**)—DDDR programming with LV-only pacing in optimal fusion intervals in an HCM patient with LBBB; (**E**)—DDD programming with biventricular pacing in an HCM patient; (**F**)—post-procedural radiological aspect of CRT-D device with mid-septal position of single-coil RV lead and posterolateral position of LV bipolar lead in a non-dilated hypokinetic HCM patient; (**G**)—post-procedural radiological aspect of a preexisting single-chamber ICD upgraded to CRT-D with a mid-septal position of dual-coil RV lead and posterolateral position of LV tetrapolar lead in a patient with end-stage dilated phenotype of HCM with severe LV dysfunction with newly developed LBBB; (**H**)—post-procedural radiological CRT aspect with mid-septal position of single-coil RV lead and posterolateral position tetrapolar multipoint LV lead in a patient with prior surgical septal myectomy. Abbreviations: LBBB, left bundle branch block; CRT, cardiac resynchronization therapy; HCM, hypertrophic cardiomyopathy; RV, right ventricle; LV, left ventricle.

**Figure 6 biomedicines-11-00350-f006:**
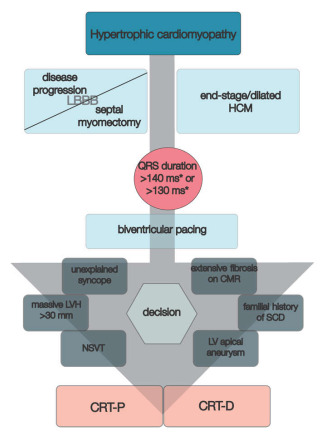
CRT in primary prevention of SCD in patients with hypertrophic cardiomyopathy. * > 140 ms in men and >130 ms in women. Abbreviations: CMR, cardiac magnetic resonance imaging; CRT-D, cardiac resynchronization therapy with defibrillator; CRT-P, cardiac resynchronization therapy with pacemaker; HCM, hypertrophic cardiomyopathy; LV, left ventricle; LVH, left ventricle hypertrophy; NSVT, non-sustained ventricular tachycardia; SCD, sudden cardiac death.

**Table 1 biomedicines-11-00350-t001:** Multimodal imaging findings in patients with hypertrophic cardiomyopathy.

IMAGING METHOD	FINDINGS	DETAILS
**ECHOCARDIOGRAPHY**	LV hypertrophy	LV wall thickness of at least 15 mmLV wall thickness of 13-14 mm in patients with familial history of HCM IVS/ILW > 1.3
	LVOT dynamic obstruction	LVOT pressure gradient of at least 30 mmHg at rest/ Valsalva maneuver/standing/exertion LVOT pressure gradient of over 50 mmHg suggests hemodynamically relevant LVOT dynamic obstruction
	SAM	Systolic aspiration of the anterior leaflet into the LVOT, leading to coaptation deficiency
	LV strain	Impaired GLS
	MV apparatus	Anterior or posterior leaflet elongation Papillary muscle abnormalities
**CMR**	Cine SSFP CMR	Chamber quantification LV wall thickness: basal asymmetrical septal hypertrophy; midventricular obstruction; apical HCM; combined hypertrophy; RV hypertrophy Papillary muscle abnormalitiesBasal crypt and apical pouching Basal crypt and apical pouching
	LGE	Myocardial irreversible replacement fibrosis: amount, distribution/pattern, localization
	T2-weighted	Myocardial edema
	PC CMR	Quantification of the blood passing through LVOT
	T1-mapping	Diffuse myocardial fibrosis

Abbreviations: Cine SSFP CMR, cine steady-state free precession cardiac magnetic resonance imaging; CMR, cardiac magnetic resonance imaging; GLS, global longitudinal strain; HCM, hypertrophic cardiomyopathy; ILW, infero-lateral wall; IVS, interventricular septum; LGE, late gadolinium enhancement; LV, left ventricle; LVOT, left ventricle outflow tract; PC CMR, two-dimensional phase-contrast cardiac magnetic resonance imaging; RV, right ventricle; SAM, systolic anterior movement of the mitral valve.

**Table 2 biomedicines-11-00350-t002:** CRT in major clinical trials.

CLINICAL TRIAL	*n*	YEAR	NYHA CLASS	QRS (MS)	LVEF
**MUSTIC-SR**	58	2001	III	>150	<35%
**MIRACLE**	453	2002	III, IV	>130	<35%
**MUSTIC-AF**	43	2002	III	>200	<35%
**PATH-CHF**	41	2002	III, IV	>120	<35%
**MIRACLE ICD**	369	2003	III, IV	>130	<35%
**CONTAK CD**	227	2003	III, IV	>120	<35%
**MIRACLE ICD II**	186	2004	II	>130	<35%
**PATH-CHF II**	89	2004	III, IV	>120	<35%
**COMPANION**	1520	2005	III, IV	>120	<35%
**CARE-HF**	814	2006	III, IV	>120	<35%
**REVERSE**	610	2008	I, II	>120	<40%
**MADIT-CRT**	1800	2009	I, II	>130	<30%
**RAFT**	1798	2010	I, II	>130	<30%

Abbreviations: LVEF, left ventricle ejection fraction; *n*, number of patients; NYHA, New York Heart Association.

## Data Availability

Not applicable.

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
