# Peer review of "Cardiac Resynchronization Therapy and Hypertrophic Cardiomyopathy: A Comprehensive Review"

_biomedicines, 2023, doi:10.3390/biomedicines11020350_

Round 1
Reviewer 1 Report
This review considers the utility and indications for cardiac resynchronisation therapy in patients with hypertrophic cardiomyopathy. The review makes clear that this is a controversial area of treatment, since clinical parameters for patient selection is relatively poorly defined, and there is a risk of doing harm in some patients, primarily by worsening the extent of heart failure. The authors also make the case that early intervention prior to significant heart failure may be beneficial, but again more research is required in this area to more clearly define the CM patient subgroups that would benefit. Overall, this timely review provides a clinically useful summary of our current knowledge in this area.
The paper is well written, with a lengthy background to HCM section that clarifies some of the difficulties of genotype-phenotype correlation, particularly in relation to the development of arrhythmias. Tables are presented that summarise the imaging findings in these patients na figures illustrate the imaging outcomes very well.
There are a small number of very minor English language issues that the authors may wish to consider prior to their final submission:
1. Page 1, abstract, line 6: “Several HCM patients are AT RISK OF DEVELOPING LBBB because of…”
2. P1, Background, L2: “….myocardial disease characterized BY asymmetrical left ventricle…”
3. P1, Background, L5: “….evolves with A poor prognosis…”
4. P2, para 2, L3: “….due to RELIEF OF symptoms….”
5. P2, para 2, L8: “…are required to enlighten OUR UNDERSTANDING.”
6. P3, para 1, L2: “…differ significantly between CAUSATIVE genes…”
7. P3, para 2, L1: “…several clinical hallmarks are OFTEN present in patients….”
8. P3, para 4, L1: “On the other HAND, advanced multimodal imaging…..”
9. P4, Table 1 list of abbreviations: Add: LGE, late gadolinium enhancement
10. P8, para 1, L1: “What is more, due to its increased risk of SUDDEN CARDIAC DEATH (SCD), over the time….”
11. P9, para 3, line 1: “As HCM progresses, the myocardium becomes more fibrotic and electricalLY hetero[1]genous….”
12. P9, para 3, L6: “….dysfunction and remodeling, SUBSTANTIALLY impacting LV systolic….”
13. P10, para 1, L2: “…QRS interval have been PREMATURELY ended because….”
14. P10, para 2, L3: “Nevertheless, the major SHORTCOMING OF surface ECG is “
15. P12, para 4, L11: “…concluded that symptoms RELIEF was rather a placebo effect…”
16. P12, para 5, L1: “Some patients with HCM who survive IN THE LONG TERM DEVELOP HF with decreased…”
17. P14, para 2, L7: “…end-stage/dilated HCM are IDENTIFIED LATE IN THEIR DISEASE COURSE when their LVEF…”
18. P14, para 2, L9: “….present with symptoms RELIEF, increased exertion….”
Author Response
Dear Reviewer,
Thank you for your kind and useful recommendations.
We have made the suggested corrections, as they can be seen in the revised manuscript
Best regards,
Reviewer 2 Report
Interesting, current topic.
Clear, concise, well structured article
Essential aspects highlighted by this manuscript:
- advanced multimodal imaging plays a paramount role in diagnosing and risk stratifying of patients with HCM, especially by the joint use of echocardiography and cardiac magnetic resonance imaging - is an interesting, well established element of the article.
- Distinguishing patients with HCM who might benefit from CRT
- Electrical parameters in stratifying patients with HCM for CRT.
But,
the section 3 refers more to the imaging approach of HCM than to the clinical one, so it would be better to add it in the subtitle as well ”clinical and imaging approach”
In figure 6 – QRS duration 120 msec is too short. How was it established? and maybe to add His and LBBB bundle pacing.
Figure 6. "CRT in primary prevention of SCD in patients with hypertrophic cardiomyopathy" - to be reformulated with Algorithm for selection of patients with CMH for biventricular stimulation. CRT is not indicated in the prevention of SCD.
Section 7 - Some data on how myomectomy influences the risk of sudden death would be interesting.

Author Response
Dear Reviewer,
Thank you for your kind feedback and for noticing these shortcomings. We have made the recommended modifications, as suggested.
- The title for Section 3 was changed from "Clinical approach in HCM" to "Clinical and imaging approach in HCM"
- We wish to thank for this recommendation. This was a gap. We changed to >140 ms in men and >130 ms in women. Regarding His and left bundle pacing, according to few available studies and to our experience as a tertiary centre in Electrophysiology, Arhythmology and Cardiac Pacing, pacients with HCM who underwent these types of cardiac pacing had a negative short clinical evolution and prognostic. Thus, we wish, with this Figure to provide an useful clinical recommendation and we wish not to add them to the figure.
- As recommended, we have added specific data in accordance with the subject of the review "More than that, supplementary attention should be given to patients with obstructive HCM who underwent septal myomectomy. In the prospective study of Woo et al [71] which evaluated over 330 patients with HCM and myectomy, at follow-up the incidence of sudden cardiac death was under 4%, while in another similar study conducted by Ommen et al [72], the 10-year survival was around 99% in those who underwent myectomy. Thus, HCM patients who undergo myectomy have a significantly decreased risk of sudden cardiac death, but it is not null. Those who require CRT should also be stratified according to current international guidelines in order to decide if a CRT-D would be a better option for them. "
We hope that we've answered all your questions as expected.
Thank you!